# *Cunninghamella arunalokei* a New Species of *Cunninghamella* from India Causing Disease in an Immunocompetent Individual

**DOI:** 10.3390/jof7080670

**Published:** 2021-08-19

**Authors:** Vinaykumar Hallur, Hariprasath Prakash, Mukund Sable, Chappity Preetam, Prashanth Purushotham, Rabindra Senapati, Shamanth Adekhandi Shankarnarayan, Nerbadyswari Deep Bag, Shivaprakash Mandya Rudramurthy

**Affiliations:** 1Department of Microbiology, All India Institute of Medical Sciences, Bhubaneswar 751019, India; vinay118@gmail.com (V.H.); dr.prashanth.pdk@gmail.com (P.P.); rabindrakumarsenapati@gmail.com (R.S.); 2Medical Microbiology, Department of Public Health, International Higher School of Medicine, Issyk-Kul Regional Campus, Cholpon-Ata 722125, Kyrgyzstan; harisath2003@gmail.com; 3Department of Pathology and Laboratory Medicine, All India Institute of Medical Sciences, Bhubaneswar 751019, India; pathol_mukund@aiimsbhubaneswar.edu.in; 4ENT and Head and Neck Surgery, All India Institute of Medical Sciences, Bhubaneswar 751019, India; ent_preetam@aiimsbhubaneswar.edu.in; 5Department of Medical Microbiology, Postgraduate Institute of Medical Education and Research, Chandigarh 160012, India; shamanth.adekhandi@gmail.com; 6Department of Radiodiagnosis, All India Institute of Medical Sciences, Bhubaneswar 751019, India; radiol_nerbadyswari@aiimsbhubaneswar.edu.in

**Keywords:** *Cunninghamella*, fungal infection, human, immunocompetent host, India, mucormycosis, new species, review, rhino-orbital-cerebral mucormycosis, taxonomy

## Abstract

Mucormycosis due to *Cunninghamella* spp. is a rare disease, especially in immunocompetent individuals. Here, we describe the isolation and characterization of a new species of *Cunninghamella*, causing chronic rhino-orbital-cerebral disease, and review cases of mucormycosis due to *Cunninghamella* spp. in immunocompetent individuals. The Basic Local Alignment Search Tool (BLAST) analysis of the internal transcribed spacer region (ITS) sequence of isolate NCCPF 890012 showed 90% similarity with *Cunninghamella bigelovii*, while the large ribosomal subunit (28S) and translation elongation factor-1 alpha (EF-1 alpha) gene sequences showed 98% identity. Further, the phylogenetic analysis with concatenated sequences clustered isolate (NCCPF 890012) closely with *C. bigelovii*. The ITS sequence showed the maximum variation among three genes analyzed and helped in the new species’ delineation. Comparison of the assembled whole genome of NCCPF 890012 with other Mucorales using 123 single-copy orthologous genes showed clustering within the genus *Cunninghamella*. Based on these findings, the isolate is considered to be a new species of *Cunninghamella* and designated as *Cunninghamella arunalokei* sp. nov. Despite repeated debridement and antifungal treatment, the patient had multiple recurrences with intracranial extension and succumbed to the illness.

## 1. Introduction

Mucormycosis is an angioinvasive disease often associated with high morbidity and mortality [1]. Infections due to *Cunninghamella* species are relatively rare in India in comparison to other countries [2]. Patients with hematological malignancy and hematopoietic stem cell transplant have an increased risk of acquiring mucormycosis due to *Cunninghamella* species which predominantly presents as pulmonary and disseminated infections [3]. Previous data show that *Cunninghamella* species are associated with high mortality compared to *Rhizopus* species [3,4]. A high mortality rate is possibly due to the pathogen’s virulence compared to other Mucorales [5].

In recent years, an increasing number of new pathogenic Mucorales species have been described using molecular techniques [6]. The internal transcribed spacer (ITS) region is commonly used in the barcoding of Mucorales [6]. This region shows a high number of polymorphic sites across different species, which helps in species identification/delineation [7]. Yu et al. reported the ITS region as a reliable marker for taxonomical delineation of *Cunninghamella* species, followed by EF-1 alpha [8]. A total of 15 species and three varieties of *Cunninghamella* are accepted to date based on morphological and molecular features; of those described species, *C. echinulata* has shown a high degree of genetic variation in the ITS region [8,9]. The pathogenic species associated with human infections are *C. bertholletiae*, *C. blakesleeana*, *C. echinulata*, and *C. elegans* [6]. In the present study, we report a case of chronic invasive rhino-orbital-cerebral mucormycosis (ROCM) due to novel *Cunninghamella* species in India. The isolate was identified and described based on phenotypic and molecular methods and designated here as *Cunninghamella arunalokei* sp. nov.

## 2. Materials and Methods

### 2.1. Clinical Details and Laboratory Diagnosis

History, physical examination details, and biopsy for histopathology and microbiology workup were collected at the time of admission at All India Institute of Medical Sciences (AIIMS), Bhubaneswar, India. Hematoxylin-eosin staining was conducted to visualize fungal elements and tissue reaction. A portion of the biopsy was also inoculated on Sabourauds dextrose agar and incubated at 28 °C. DNA from formalin-fixed paraffin-embedded tissue (FFPE) was extracted using phenol: chloroform method as described previously by Jillwin et al. [10] and the fungal PCR was performed using ZM primers, as described previously [10,11].

### 2.2. Phenotypic Characterization

For morphological studies, slide culture was performed using potato dextrose agar (in house), mounted in lactophenol cotton blue on day 4 and observed under ×50 magnification with Carl Zeiss AXIOSkop with Axiocam ICc 5 camera and Zen 2.3 Software (Carl Zeiss Microscopy GmbH, Gottingen, Germany). A total of 30 consecutive sporangiophore, vesicle, sporangiola, and spores were measured, and the arithmetic mean values were determined. Micromorphological characteristics were studied by scanning electron microscopy (JSM-IT300, JEOL, USA) following the protocol described by Alves et al. [12]. Experiments to determine physiological properties like growth at different temperatures (25 °C, 28 °C, 32 °C, 35 °C, 37 °C, 39 °C, 40 °C, and 42 °C), different pH (5, 7, 9, 11) and salt concentrations (0%, 1.5%, 2%) were performed in duplicate.

### 2.3. Antifungal Susceptibility Testing

The antifungal susceptibility testing of *Cunninghamella* strain NCCPF 890012 was performed following CLSI guidelines M38-A2. The minimum inhibitory concentration (MIC) was measured for the following antifungals: amphotericin B, itraconazole, posaconazole, voriconazole, caspofungin, anidulafungin, and micafungin.

### 2.4. Molecular Characterization

Genomic DNA from the mold was extracted using the phenol chloroform isoamyl alcohol method described previously and resuspended in TE buffer (10 mM Tris-HCl, 1 mM EDTA [pH 8.0]). The ITS, 28S, and EF1-alpha genes were amplified by polymerase chain reaction (PCR) using the respective primers; ITS-5(5′-GGAAGTAAAAGTCGTAACAAGG-3′) and ITS-4 (5′-TCCTCCGCTTATTGATATGC 3′) for ITS region, NL-1 (5′-GCATATCAATAAGCGGAGGAAAAG-3′) and NL-4 (5′-GGT CCGTGTTTCAAGACGG-3′) for 28S, and EF-1F (5′-ACAACAGGCTGGTATCTC-3′) and EF-1R (5′-AACCTTACCGGCCTTATC-3′ for EF-1 alpha. For ITS and 28S genes, the PCR was performed as previously described by Prakash et al. [13]. For EF1-alpha, the following amplification protocol was followed: initial denaturation at 94 °C for 1 min followed by 35 cycles at 94 °C for 1 min, 56 °C for 1 min and 72 °C for 2 min. Sequencing was performed using the primers mentioned above and a BigDye Terminator cycle sequencing kit (version 3.1; Applied Biosystems, Foster City, CA, USA). Sequencing reactions were analyzed on an ABI 3500 genetic analyzer (Applied Biosystems). Consensus sequences obtained using Bionumerics software (version 7.2; Applied Maths, Ghent, Belgium) were compared with the GenBank (Available online: http://www.ncbi.nlm.nih.gov/GenBank/index.html (accessed on 20 December 2020) and CBS-KNAW Fungal Biodiversity Centre (Available online: http://www.cbs.knaw.nl (accessed on 20 December 2020) database.

### 2.5. Phylogenetic Tree Construction

For phylogenetic analysis of the *Cunninghamella* species, the nucleotide sequences of the following genes were retrieved from NCBI (Available online: https://www.ncbi.nlm.nih.gov/ (accessed on 5 January 2021); (a) internal transcribed spacer (ITS) region (b) large ribosomal subunit (28S); and (c) translation elongation factor-1 alpha (EF-1alpha). The accession numbers are given in Appendix A. The phylogenetic tree for each gene was constructed as follows: the sequences were subjected to multiple sequence alignment using Clustal X software [14], then, phylogenetic trees were built for these genes using the Maximum Likelihood method and Kimura 2-parameter model by using Molecular Evolutionary Genetics Analysis (MEGA 7) software [15]. The super-tree was constructed using the phylogenetic tree of the individual genes (ITS, 28S and EF-1alpha) using CLANN software [16].

### 2.6. Whole-Genome Phylogenetic Analysis

A total of 56 genomes were shortlisted for the whole-genome phylogenetic analysis. A total of 41 assembled genomes were downloaded from NCBI, and for 14 genomes, raw data were downloaded from NCBI and assembled by SPAdes [17]. The genome of *Cunninghamella arunalokei* sp. nov. was sequenced by Illumina and assembled by SPAdes in this study (Appendix A). These genomes were subjected to BUSCO analyses to identify proteins that are commonly present in all the above. Multiple sequence alignment was performed [18] for these protein sequences, and poorly aligned regions were trimmed using trimAl [19]. Then these sequences were concatenated together, and a maximum-likelihood phylogenetic tree was constructed by the IQ-TREE approach [20].

### 2.7. Ethical Statement

This study with project code T/IM-NF/Micro/21/42 was approved on 25 June 2021 by the Institutional Ethics Committee, All India Institute of Medical Sciences Bhubaneswar.

## 3. Results

### 3.1. Clinical Details and Laboratory Diagnosis

A 26-year-old Indian male from Odisha presented to us with erythema, induration, and superficial ulceration with crusting on the left dorsum of the nose and swelling in the left mid-half of the face. On inquiry, he informed that he had a history of cheek swelling with ulceration of the overlying skin for the last 6 years, for which he consulted a surgeon elsewhere. Histopathological examination report of the biopsied lesion and radiological imaging had confirmed the diagnosis of invasive mucormycosis with involvement of the facial skin, subcutaneous tissue, and the maxillary sinuses. He had undergone debridement of the skin lesions and the maxillary sinus cavity and had received 3 g of liposomal amphotericin B. However, the next 2 years, he suffered from recurrences and underwent repeated surgeries and intermittent antifungal therapy with liposomal amphotericin B, posaconazole alone, and a combination of posaconazole and caspofungin. Serial imaging revealed progressive disease. Failing to get any relief from the disease, he reported to AIIMS, Bhubaneswar. On presentation, his left ala was destroyed, exposing the nasal septum (Figure 1A). His laboratory investigations revealed normal hemogram, blood sugar levels, liver, and renal function parameters. Urine ketone bodies were not detected. Serological examination for HIV was non-reactive, and the lymphocyte CD4 counts were in the normal range. His immunoglobulin and complement levels were also within normal limits. The lesions were biopsied again, and he was started on a combination of super saturated potassium iodide solution (SSKI) and oral itraconazole, considering a probability of atypical entomophthoramycosis. A potassium hydroxide mount of the biopsy showed broad aseptate hyphae (Figure 2A), and culture grew a white cottony mold at 28 °C after 5 days of incubation. Lactophenol cotton blue examination of the mold demonstrated erect sporangiophores ending in a swollen vesicle with 1-spored sporangiola suggestive of *Cunninghamella* species. Histopathological examination from the biopsied nasal mucosa showed fibro collagenous tissue with non-necrotizing granulomatous inflammation with florid multinucleated foreign body giant cell reaction and intracellular and extracellular broad aseptate hyphae (Figure 2B). Few of the small vessels showed the presence of fungal profiles with the destruction of the vessel wall. In addition, intra and perineural invasion was also noted. PCR and sequencing of the formalin-fixed paraffin-embedded tissue using the ZM primers also confirmed the presence of *Cunninghamella* spp. Based on the above findings, a diagnosis of chronic invasive mucormycosis with recurrence was established, and treatment with SSKI and itraconazole was stopped. The lesion was debrided repeatedly, and terbinafine was administered as the patient had never received terbinafine in the past. Terbinafine was continued for 3 months, after which the facial lesions healed (Figure 1). However, 5 months after the cessation of medication, he again presented with recurrent epistaxis and decreased facial sensation. A fresh biopsy confirmed a disease recurrence, and the patient was treated with a combination of oral posaconazole and terbinafine for 6 months. The lesions regressed, and he was advised to continue oral terbinafine indefinitely. However, the patient could not procure terbinafine during COVID-19 pandemic lockdown for around 7 months and reported 2 months later to the emergency with disorientation, severe weakness due to reduced oral intake secondary to palatal fistula (Figure 1), and nasal regurgitation. The facial swelling further increased and was associated with crusting (Figure 1). A repeat biopsy culture performed confirmed recurrence. Magnetic resonance imaging revealed disease extension to anterior cranial fossa with skull base erosion (Appendix A). The patient was initiated on liposomal amphotericin B. A total of 4 g of liposomal amphotericin B was administered during four weeks. The patient improved gradually, and the lesion subsided. The patient was discharged on oral posaconazole 300 mg/day. However, the patient did not take the posaconazole due to financial constraints and succumbed to the illness 1 month after discharge.

### 3.2. Phylogenetic Analysis

The NCBI accession numbers of the *C. arunalokei* sequences are: (a) ITS (MN431159.1); (b) 28S (MN431158.1) and (c) EF-1 alpha (MZ821065).BLAST search of the ITS sequence of NCCPF 890012 showed 97% identity with an undescribed *Cunninghamella* species (accession no: MG571234.1), and the 28S and EF-1 alpha sequences showed 98% identity with *Cunninghamella bigelovii* (accession no of 28S and EF-1 alpha genes respectively: KJ013405.1 and KJ395944.1). The BLAST search of the ITS sequence showed 90% identity (72 variable sites in 736 total number of sites analyzed) with *C. bigelovii*. The individual phylogenetic gene trees (Appendix A) and the super-tree (constructed using 3 genes) clustered strain NCCPF 890012 closely with *C. bigelovii* (Figure 3) with maximum gene nucleotide variation in the ITS sequence. Based on these findings, the strain, NCCPF 890012, is considered to be a new species in the genus *Cunninghamella*, and the isolate is designated as *Cunninghamella arunalokei* sp. nov. Further, we sequenced and assembled *C. arunalokei* genome and constructed the whole-genome phylogenetic tree with the previously available genome sequences of Mucorales. A total of 123 common single-copy genes were identified using BUSCO analysis. The phylogenetic analysis using IQ-tree analysis clustered the *C. arunalokei* within the genus *Cunninghamella* (Figure 4).

### 3.3. Antifungal Susceptibilit

The isolate showed MICs of 1 µg/mL to amphotericin B, and terbinafine, 4 µg/mL to posaconazole, 8 µg/mL to voriconazole, caspofungin, anidulafungin, and micafungin.

### 3.4. Taxonomy

*Cunninghamella arunalokei* V. Hallur, S. Rudramurthy H. Prakash, sp. nov.

MycoBank MB840808

Most of the sporangiola of this species were globose in shape and predominantly echinulate.

Type: Isolated from sinonasal tissue of a patient with chronic mucormycosis on 12 March 2018, V. Hallur, S. Rudramurthy and H. Prakash (Holotype: dried culture at the NCCPF, Chandigarh, India (*C. arunalokei* NCCPF 890012). Living ex-type culture NCCPF 890012.

Etymology: *Cunninghamella arunalokei*, the species epithet arunalokei (a.ru.na.lo’ke.i. NL masc. gen. n. arunalokei, of Arunalokei) in honor of Arunaloke Chakrabarti, for his numerous contributions to the development of medical mycology in India and Asia.

Colonies exhibited rapid growth on Sabourauds Dextrose Agar (SDA) and Potato Dextrose Agar (PDA), attaining a diameter of 70–76 mm after 4 days of incubation at 25 °C. Colonies were initially white later turned greyish. The colonies developed scattered brown specks at ten days’ incubation. The colony on the reverse was uniformly pale grey. Sporangiophores were erect, measured 6.5–8 μm, irregularly branched and exhibited cymose pattern (Figure 5A). Sporangiola were globose and measured 8.5 ± 1 μm (Figure 5A). Hyaline to brown spores was predominantly globose echinulate spores of the same size were found (Figure 5B). Terminal vesicles were globose, 17–37 μm (24.9 ± 6.0 μm) in diameter. Lateral vesicles were the same as terminal vesicles and measured up to 23 μm in diameter. Chlamydospores and zygospores were not observed. Scanning electron microscopy of echinulate spores are shown in Figure 5C,D.

The isolate showed growth at pH between 5 and 11 and tolerated salt concentrations of up to 2%. *Cunninghamella arunalokei* isolate did not grow at or above 42 °C but grew well at 37 °C on MEA and SDA. Antifungal susceptibility testing of the isolate by microbroth dilution method (CLSI M38, A2.) exhibited minimum inhibitory concentration (MIC) levels of 1 µg/mL for amphotericin B and terbinafine, 4 µg/mL to posaconazole, 8 µg/mL to voriconazole, caspofungin, anidulafungin, and micafungin. The isolate was deposited at the National Culture Collection for Pathogenic Fungi (NCCPF), Chandigarh, India, with accession number NCCPF 890012. The isolate from the recurrence biopsy specimen showed growth of *Cunninghamella* species, with similar phenotypic and genotypic features to that of the first isolate. The isolate showed 100% nucleotide similarity in the ITS region with the initial isolate.

## 4. Discussion

*Cunninghamella* species rarely cause invasive mucormycosis, and the infections had been described predominantly in immunocompromised patients [1]. Isolation of *Cunninghamella* species in an immunocompetent host is extremely rare, and to date, only three cases have been described (see Table 1). In the present study, we report discovery of novel fungus in a case of invasive rhino-orbito-cerebral mucormycosis in an apparently immunocompetent patient. The patient presented with recurrent disease despite repeated surgical debridement and courses of antifungal therapy. Further, the possibility of increased virulence of the pathogen cannot be ruled out.

*Cunninghamella* species are saprophytic fungi, predominantly seen in soil and air samples of India [1,13]. To date, 15 species and three varieties of *Cunninghamella* have been described, of these, species reported in humans are *C. bertholletiae*, *C. blakesleeana*, *C. echinulata*, and *C. elegans*. *Cunninghamella* infections are predominantly seen in patients with hematological malignancy and hematopoietic stem cell transplant. Table 2 shows the features to differentiate clinically relevant *Cunninghamella* species. Infections due to *Cunninghamella* species in immunocompetent host is a rare entity. So far, *C. bertholletiae* is the only species in the genus *Cunninghamella* that has been associated with infections in immunocompetent host. In the present study, we report the identification of a novel species *C. arunalokei* which caused granulomatous invasive mucormycosis in an immunocompetent host. Granulomas are found infrequently in mucormycosis except in patients with chronic granulomatous disease [24,25]. Goel et al. reported that patients with a florid granulomatous response and mild angioinvasion had a better prognosis compared to the ones with mild granulomas and marked angioinvasion on histology. However, the finding was not statistically significant [26]. A marked granulomatous response with mild angioinvasion in our patient may explain the chronic nature of the infection which however was relentless and lead to the death of the patient.

The phylogenetic delineation of *Cunninghamella* species has not been studied in greater detail. As the taxonomical classification of *Cunninghamella* species solely relies on ITS barcoding, the species’ proposed to date were identified based on variation in the ITS sequences, followed by EF-1 alpha gene [6]. Unlike, ITS gene sequences, EF-1 alpha failed to show maximum sequence variation. Further, only partial gene sequences of EF-1 alpha are available in the public database which makes them a lesser important in taxonomical delineation of *Cunninghamella* species. In the present study, we identified a novel *Cunninghamella* species based on phenotypic and genotypic features. The sequences of ITS and 28S rDNA failed to show identical matches on BLAST analysis. Our phylogenetic analysis of the ITS, 28S, and EF-1 alpha showed clear species delineation without any match with the previously described species. The present isolate, *C. arunalokei* clustered closely with *C. bigelovii*. Of those three genes, the ITS region showed maximum nucleotide variation (10%), followed by 28s (2%) and EF-1 alpha (2%) compared to *C. bigelovii*. Further, we attempted whole genome-based phylogeny for the phylogenetic inference of the proposed new species, *C. arunalokei* was compared with other Mucorales. The phylogenetic analysis clustered *C. arunalokei* within the genus *Cunninghamella*. In the present study, we could use three species of *Cunninghamella* in genome-based phylogeny, as only these are the available sequences in the public database.

The fungus isolated in the present study is closely related to *C. bigelovii*, an endophytic fungus isolated from the shoots of the *Salicornia bigelovii* plant [30]. While, it showed similar morphology and growth temperatures as that of *C. bigelovii*, it can be phenotypically distinguished from *C. bigelovii* based on smaller and globose sporangiola (8.5 ± 1 µm). The sporangiola of the latter are larger and ellipsoidal to subglobose with a diameter of 13 × 9 μm. Further, no branches arose from the vesicles of *C. arunalokei,* whereas branches were noted on the vesicles in *C. bigelovii* [30].

## 5. Conclusions

We report isolation of *Cunninghamella arunalokei* sp. nov. from a 26-year-old immunocompetent Indian patient with recurrent chronic invasive mucormycosis. The isolate was clinically significant as evidenced by histopathology, and isolation from biopsy sample. Our findings are unique as *Cunninghamella* rarely causes mucormycosis in immunocompetent individuals and phylogenetic analysis confirmed that it is a new species. It was named *Cunninghamella arunalokei* in the honor of Prof. Arunaloke Chakrabarti a leading clinical mycologist from India.

## Figures and Tables

**Figure 1 jof-07-00670-f001:**
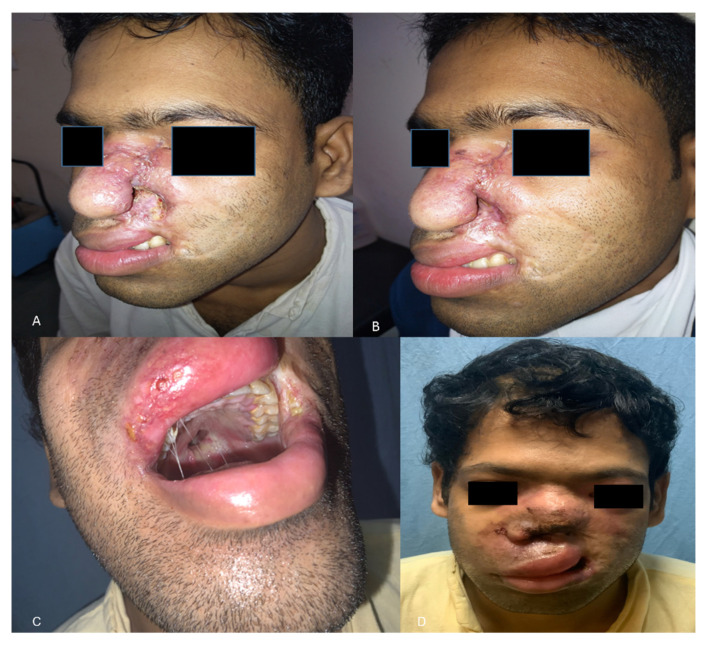
Photograph of patient (**A**) on first admission and (**B**) after 3 months of treatment with terbinafine (**C**) on second admission with palatal fistula and (**D**) recurrent facial lesions.

**Figure 2 jof-07-00670-f002:**
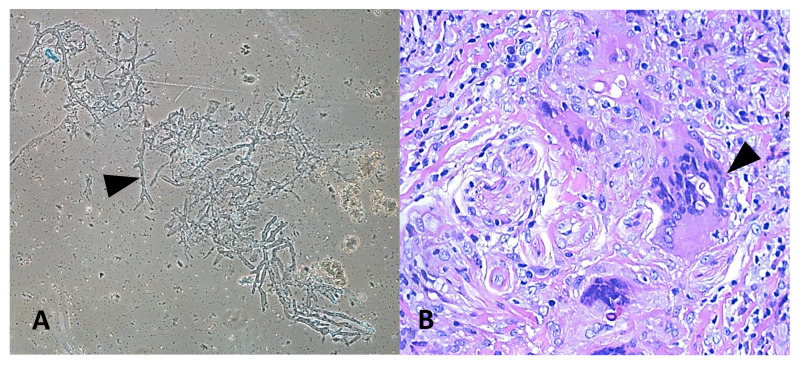
(**A**) Potassium hydroxide mount of nasal scraping showing broad aseptate hyphae (indicated by triangle) (10×) (**B**) histopathological findings of biopsy from exposed nasal mucosa showing non-necrotizing granulomas with broad aseptate fungal profiles (indicated by triangle).

**Figure 3 jof-07-00670-f003:**
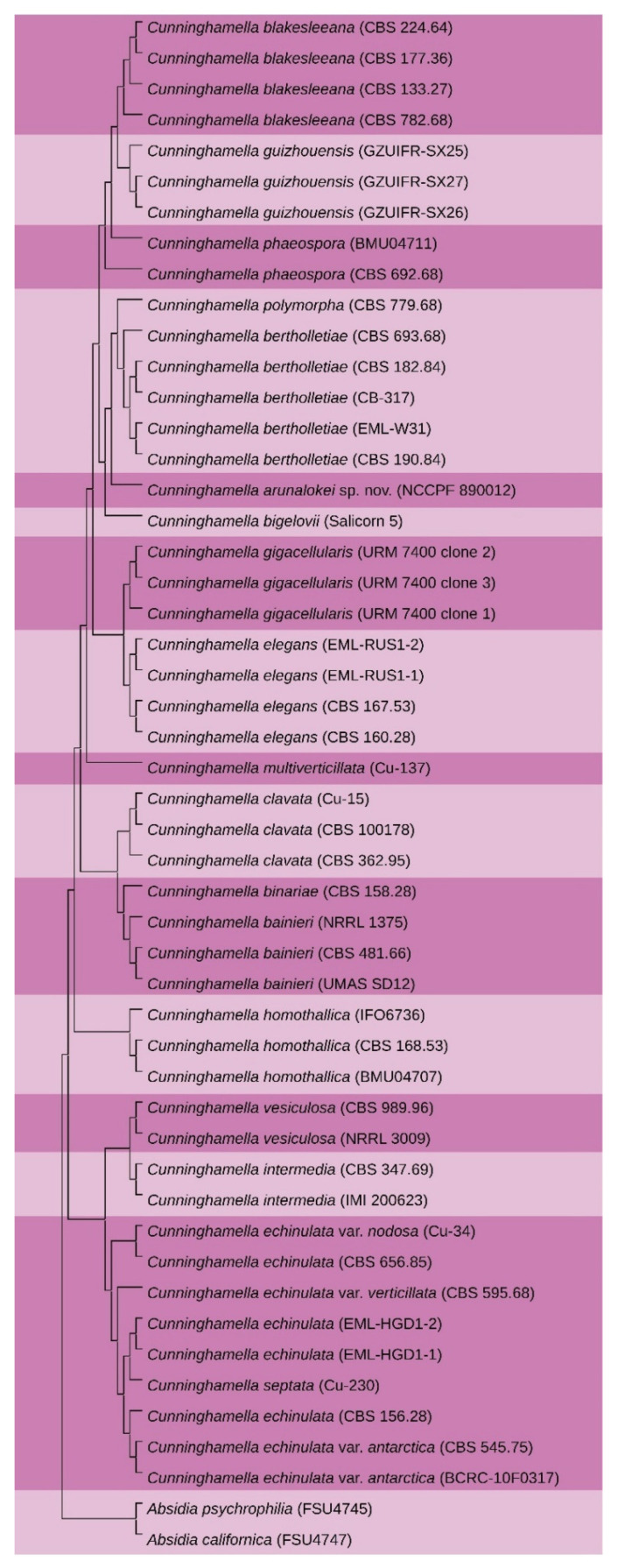
Super-tree constructed using three genes i.e., ITS, 28s, and EF-1 clustered isolate *Cunninghamella arunalokei* (NCCPF 890012) closely with *C. bigelovii*.

**Figure 4 jof-07-00670-f004:**
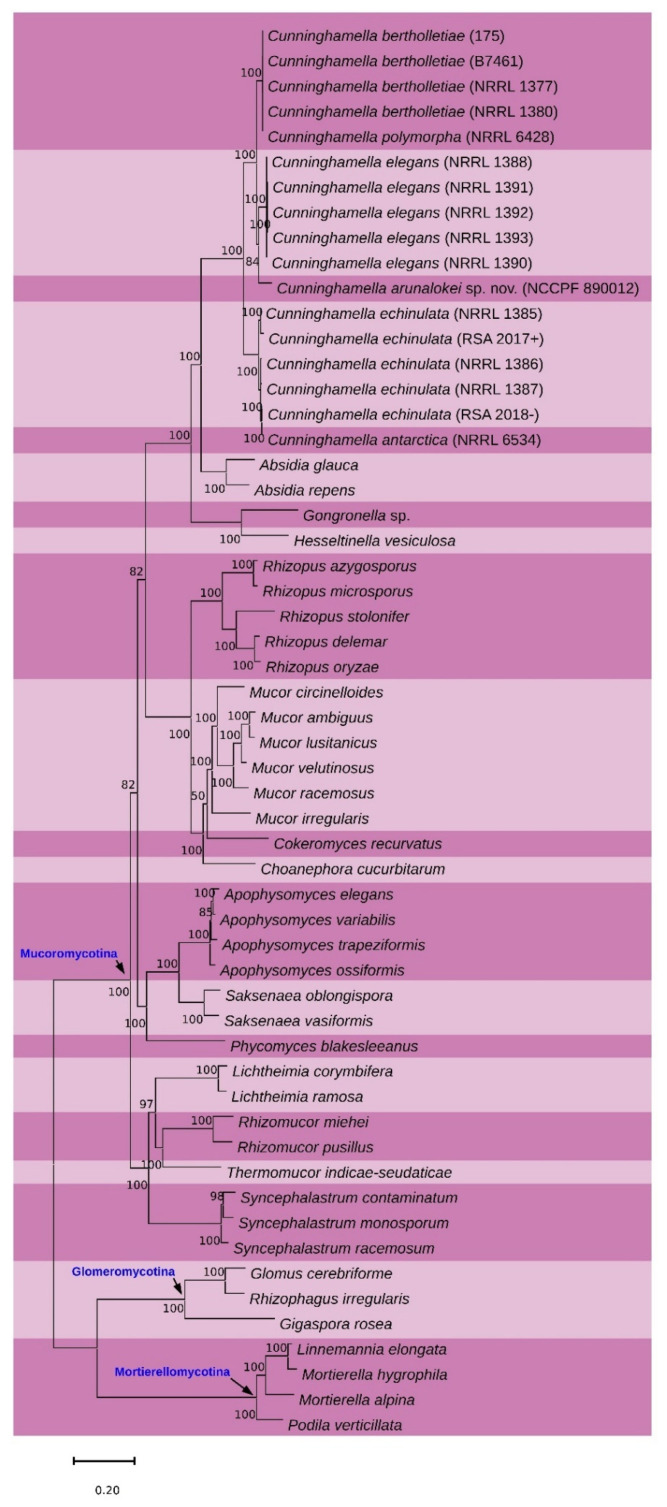
Whole genome phylogeny of *Cunninghamella arunalokei*.

**Figure 5 jof-07-00670-f005:**
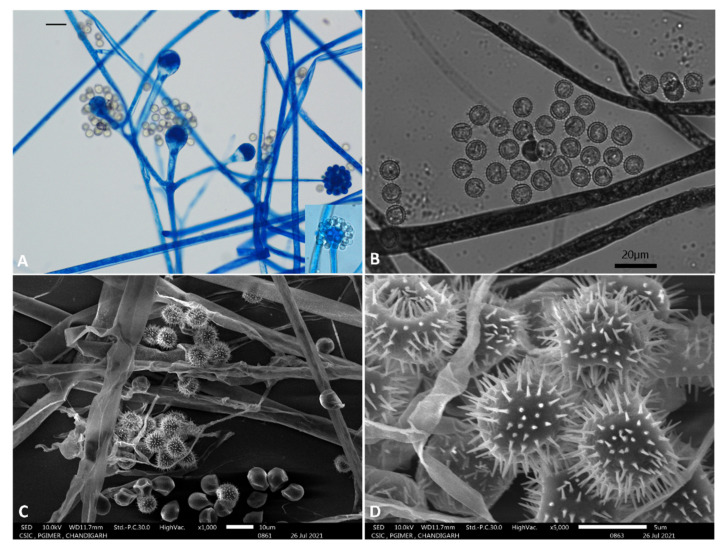
(**A**) Bright field microscopy showing mold with erect sporangiophore ending in a swollen vesicle, with 1-spored sporangiola(inset) and lateral branches with vesicles and sporangiola with a whorl of 3–5 branches suggestive of *Cunninghamella* (Scale bar 20 µm) (**B**) Brightfield microscopy showing echinulate spores (Scale bar 20 µm). (**C**) Scanning electron microscopy of *Cunninghamella arunalokei* showing rough sporangiola, some of which are collapsed (Scale bar 10 µm). (**D**) Scanning electron microscopy of *Cunninghamella arunalokei* showing echinulate spores (Scale bar 5 µm).

**Table 1 jof-07-00670-t001:** Salient clinical features of *Cunninghamella* species reported from immunocompetent individuals.

Country	Age	Sex	System Involved	Treatment	Clinical Presentation	Species	Outcome
USA [21]	61	Male	Lungs	Amphotericin B	Invasive pulmonary mucormycosis	*C. bertholletiae*	Death
Sri Lanka [22]	42	Male	Sinus and face	Intermittent Amphotericin B	Rhinofacial invasive mucormycosis	*Cunninghamella* sp.	Cured
Japan [23]	74	Male	Lungs	Voriconazole + Amphotericin B	Invasive pulmonary mucormycosis	*C. bertholletiae*	Death
Present case	26	Male	Face, sinus, and brain	Amphotericin B, Posaconazole, Caspofungin, Terbinafine and Potassium Iodide	Rhinofacial chronic granulamatous invasive mucormycosis	*C. arunalokei*	Death

**Table 2 jof-07-00670-t002:** Showing phenotypic characters to differentiate clinically relevant *Cunninghamella* species.

Name	Colony Color At Maturity	Growth at 40 °C	Terminal Vesicles	Sporangiola	Differentiating Feature
*C. arunalokei*	Grey	Restricted	Globose	Translucent to brown, globose, and echinulate	Restricted growth at 40 °C with grey colonies at maturity and presence of brown sporangiola on vesicles ≤ 30 µm [8]
*C. bertholletiae*	Grey	Good	Subglobose to obovate	Translucent to brown, spherical to ellipsoid, rarely tear drop shaped, smooth to punctate	Sporangiophores branching pattern is diverse and sporangiola are tear shaped [8,27]
*C. blakesleeana*	White to light buff	None	Spherical to obovate	Translucent, spherical to ellipsoidal, smooth to echinulate with spines up to 1.6 µm	Opposing suspensors of the zygospores are unequal in size [28]. Finely echinulate to smooth sporangiola borne on solitary or verticillate branches of sporangiophores are reliable character [27].
*C. echinulata*	White to pale buff	Good	Spherical to subglobose	Translucent, mainly spherical occasionally subglobose to broadly ellipsoidal, studded with spines up to 4 µm	Growth has powdery appearance at 40 °C and sporangiophores branch in pseudo verticillate manner [28]
*C. elegans*	Grey	None	Subglobose to broadly clavate	Translucent to pale brown, spherical to ellipsoidal, smooth to echinulate with spines up to 0.8 µm	Very similar to *C. bertholletiae* but can be differentiated by absence of growth at 40 °C [29]

## Data Availability

The data presented in this study are available in the article or Appendix A. The genome data of *Cunninghamella arunalokei* generated in this study will be provided on request, contact Dr. Vinaykumar Hallur.

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
