# Peer review of "Cunninghamella arunalokei a New Species of Cunninghamella from India Causing Disease in an Immunocompetent Individual"

_jof, 2021, doi:10.3390/jof7080670_

Round 1

Reviewer 1 Report

Three files are attached, two corresponding to the manuscript and the complementary material, both with some comments. The third file is an article describing the new species Cunninghamella bigelovii, which should be taken as a model to validly describe the new species. 

Author Response

Response to comments in the main manuscript

Point 1: Strikethrough text on line 17

Response to 1: , has been striked off

Point 2: Add and in line 17

Response to 2: Added “and” in line 17

Point 3 Add , in line 18

Response to point 3: Added , in line 18

Point 4 The picture C appears to be deformed compared to the rest.

Response to point 4: This is the only photograph available with us. However, we have modified and adjusted the size again to fit the frame.

Point 5 Check if the text of the Figure is in accordance with the editorial standards.

Response to point 5: Changed the figure legend to  “Figure 1.  Photograph of patient A) on first admission and B) after three months of treatment with terbinafine C) on second admission with palatal fistula and D) recurrent facial lesions.” in lines 200-202

Point 6 Poor resolution. I don't sure if the Figure has the right position in the page.

Response to point 6: The combined image has been split into two separate figures which has improved the resolution. The final typesetting can be done by professionals of Journal of fungi at  Lines 230 & 235

Point 7 Description of the new species is not according to the International Code of Nomenclature for Algae, Fungi, and Plants (https://www.iapt-taxon.org/nomen/main.php) and, consequently, should be invalidated. Please, read carefully the article "Isolation of Cunninghamella bigelovii sp.nov. CGMCC 8094 as a new endophytic oleaginous fungus from Salicornia bigelovii", because it is an excelent example of description of a new species belonging to the same genus.

Response to point 7: The description of the fungus has been changed to “

Cunninghamella arunalokei V. Hallur, S. Rudramurthy H. Prakash , sp. nov.

Most of the sporangiola of this species were globose in shape and predominantly echinulate.

Type Isolated from sinonasal tissue of a patient with chronic mucormycosis on 12th March 2018, V. Hallur  S. Rudramurthy and H. Prakash (Holotype: dried culture at the NCCPF, Chandigarh, India (C. arunalokei NCCPF 890012). Living ex-type culture NCCPF 890012” in lines 240 to 246

Point 8 Change C. arunalokei to Cunninghamella arunalokei in line  232

Response to point 8: Has been changed to Cunninghamella arunalokei at line 263

Point 9 Image quality is unacceptable and non-representative, becasue micrographs taken by stereo- and bright field microscopes are at a very low magnification, and SEM micrographs shown only collapsed structures.

Response to point 9: We tried multiple times with multiple protocol to get the best images of SEM.  New images with good resolution of both SEM and bright field microscopic pictures are provided in this revised version of the manuscript.

Point 10 to 15 Add spaces before references in the table 2 on page 14 of the document

Response to point 10 to 15 :  have been added  in the table 2 on page 14 of the document

Response to comments in Supplementary file

Point 1: These pictures must be included in the main body of the manuscript. Please, intro arrows to locate the mentioned structures in the pictures. Increase contrast picture B.

Response to point 1: The said pictures are now included in the main manuscript. Contrast of Picture B has been increased.

Point 2: Delete “Showing” from figure legend of radiological images over the years.

Response to point 2: Has been deleted from the figure legend.  The figure legend now reads “Radiological progression of the lesion over the five years A) lesion in the skin and subcutaneous tissue of left side of the face (2015) B & C) progression with involvement of the left maxillary sinus mucosa thickening (2016 & 2017) & D) Intracranial extension in 2021.”

Point 3: Poor quality. Increase it! of  Figure S3

Response to point to 3: The combined figure has now been split into three individual figures, improving the quality of the figures.

Points 4 & 5:  Redistribute the contain of the columns to decrease the size of the tables.

Response to points 4& 5: Have been redistributed to decrease the size of the tables

As advised by Ms. Nicoleta Vaniga, Assistant Editor Ethical statement has also been added to the manuscript.  

Reviewer 2 Report

The manuscript titled “Cunninghamella arunalokei a new species of Cunninghamella from India causing disease in an immunocompetent individual” is a welcome work that can contribute to studying the deadly disease mucormycosis. This work reports a new species involved in mucormycosis, showing a high virulence in an immunocompetent patient. The authors described the morphological specific features of this fungus. Moreover, the molecular analysis using the ITS, EF-1 alpha, and whole-genome sequencing helped correctly characterize this new species. I have only one major concern. The authors described the unsuccessful antifungal treatment used in the patient (although the patient did not properly follow it for several reasons). In this sense, it would be beneficial for the clinicians if the authors characterized the antifungal response of this new fungus to the different types of antifungal drugs in vitro, which would help in future cases of mucormycosis caused by Cunninghamella arunalokei.

Author Response

Point 1: The manuscript titled “Cunninghamella arunalokei a new species of Cunninghamella from India causing disease in an immunocompetent individual” is a welcome work that can contribute to studying the deadly disease mucormycosis. This work reports a new species involved in mucormycosis, showing a high virulence in an immunocompetent patient. The authors described the morphological specific features of this fungus. Moreover, the molecular analysis using the ITS, EF-1 alpha, and whole-genome sequencing helped correctly characterize this new species. I have only one major concern. The authors described the unsuccessful antifungal treatment used in the patient (although the patient did not properly follow it for several reasons). In this sense, it would be beneficial for the clinicians if the authors characterized the antifungal response of this new fungus to the different types of antifungal drugs in vitro, which would help in future cases of mucormycosis caused by Cunninghamella arunalokei.

Response to point 1: Results of antifungal susceptibility testing have been added in lines 236-238 as “The isolate showed MICs of  1µg/ml  to amphotericin B, and terbinafine, 4µg/ml to posaconazole, 8µg/ml to voriconazole, caspofungin anidulafungin, and micafungin.”

Round 2

Reviewer 1 Report

Dear authors,

All correcions and comments are in the attached version of the manuscript. It is very important to obtain a Mycobank (www.mycobank.org) accession number to validate your new species.

Best regards,

Author Response

Response to Reviewer 1 Comments

We thank the reviewer for the excellent review and inputs which have improved the scientific rigor of the article. A pointwise response to suggestions are given below

Response to comments in the main manuscript

Multiple Points: On grammatical errors   

Response to Multiple points: Changes have been made as suggested throughout the document with track changes which is uploaded for your kind perusal

Point: Arrange Keywords in alphabetical order

Response to point on Keywords: Changes made on lines 34-35

 Point: Please, clarify, because there are included pictures from the patient. Who approbal the use of these images?

Response to point on Ethics approval:  This study with project code T/IM-NF/Micro/21/42 was approved and exempted from full board review on 25.06.2021 by the Institutional Ethics Committee, All India Institute of Medical Sciences Bhubaneswar as fungal isolates were obtained during routine patient care and written consent was obtained from patient for using clinical photographs.

“The lines Ethical statement are now modified to: This study with project code T/IM-NF/Micro/21/42 was approved on 25.06.2021 by the Institutional Ethics Committee, All India Institute of Medical Sciences Bhubaneswar.” In lines 141-145

Point on:  Geographic origin of the patient

Response to point on Geographic origin of the patient: Line 148 has been modified to “ A 26-year-old Indian male from Odisha presented to us with erythema, indu” to indicate the geographic origin.

Point on Mycobank accession number : The authors must validate the taxon by introduction all data available in Mycobank, and to obtain a Mycobank accession number, which is necessary to the introduced at next of the scientific name in the manuscript

Response to point on Mycobank accession number: All data related to the above isolate has been uploaded on the mycobank database and a mycobank accession number MB840808 has been obtained, the same has been incorporated into manuscript on line 250.

Point on Submission to Westerdijk Institute Culture Collections and Herbarium. : It would be convenient for the authors to send the holotype to a collection of cultures of recognized international prestige, such as the Westerdijk Institute Culture Collections and Herbarium.

Response to point on Submission to Westerdijk Institute Culture Collections and Herbarium:

The isolate has already been submitted to National Culture Collection of pathogenic fungi(NCCPF), Postgraduate Institute of Medical Education and Research, Chandigarh, India which is an is an affiliate member of the World Federation for Culture Collections (WFCC). Cultures deposited at the NCCPF are available for study by scientists as they are at the Westerdijk Institute Culture Collections and Herbarium. Moreover, because of the COVID-19 pandemic depositing the same to Westerdijk Institute Culture Collections and Herbarium will take a lot of time. The process of deposition to Westerdijk Institute Culture Collections and Herbarium as per the suggestion of the reviewer has been initiated and would be completed after or during the process of publication of the manuscript

Point on rearrangement of tables in manuscript

Response to point on rearrangement of tables in manuscript: As advised table 1 has been shifted to results before antifungal susceptibility i.e.on lines 241 to 244 and table 2 has been shifted to results after description of the new species

Reviewer 2 Report

The authors perfectly responded to my questions.  

Author Response

There were no queries by the Reviewer 2 in round 2